# A MULTICRITERIA NEURAL NETWORK-BASED APPROACH TO ADAPTIVE INVESTMENT PORTFOLIO FORMATION

**Khudyakov Stanislav Andreevich**[*]
Barykin Sergey Evgenievich[†]
Dinets Daria Aleksandrovna[‡]

## ABSTRACT

This paper presents a methodology for forming an adaptive investment portfolio structure based on a multi-module architecture implemented using an ensemble of neural network models. The developed system comprises five specialized analytical modules, each responsible for processing a specific class of factors: macroeconomic, fundamental, technical, and credit. The central element of the architecture is a neural network model for forecasting the Bank of Russia key interest rate, which defines scenario conditions for the valuation of all asset classes. Fundamental equity analysis, technical analysis of market dynamics, credit risk assessment of debt instruments, and an adapted Markowitz portfolio optimization model are integrated within a digital twin that reconciles the outputs of individual modules. The digital twin performs the functions of coordinating and aggregating the ensemble results, identifying inconsistencies between partial recommendations, and generating a coherent investment decision. The system produces probabilistic investment recommendations, thereby enabling scenario analysis and quantitative risk assessment. The proposed approach can be used as a decision-support tool in both individual and institutional asset management, taking into account changes in the macroeconomic environment.

**Keywords:** investment portfolio, neural network models, key interest rate, fundamental analysis, technical analysis, credit risk, Markowitz model, digital twin, adaptive optimization, monetary policy.

## 1 INTRODUCTION

Forming an investment portfolio under current conditions requires taking into account a wide range of factors that affect the return, risk, and liquidity of assets. Traditionally, investment choices rely on three key approaches: fundamental, technical, and credit risk analysis. However, in portfolio management practice these approaches are often implemented in isolation, which complicates the adoption of coordinated decisions under conditions of elevated market volatility.

A specific feature of the Russian financial market is the high sensitivity of all asset classes to changes in the parameters of monetary policy. An increase in the Bank of Russia key interest rate has multifaceted effects: it leads to a revision of the discount rate and, consequently, to a decrease in the fair value of equities; it increases market volatility; and it is accompanied by a widening of spreads on corporate bonds. The lack of systemic coordination between analytical methods and the inability to promptly incorporate changes in macroeconomic conditions create uncertainty in the investment process.

---

[*]Peoples' Friendship University of Russia, Moscow, Russia. ORCID: https://orcid.org/0009-0007-6432-7888. Email: stasansk2014@gmail.com.

[†]Doctor of Economics, Professor, Peoples' Friendship University of Russia, Moscow, Russia. ORCID: https://orcid.org/0000-0002-9048-009X. Elibrary SPIN: 9382-2074. Email: sbe@list.ru.

[‡]Doctor of Economics, Associate Professor, Peoples' Friendship University of Russia, Moscow, Russia. ORCID: 0000-0001-8734-8998. Elibrary SPIN: 2607-3650. Email: dinets-da@rudn.ru.

This paper considers a multi-module asset selection system based on specialized neural network models, each responsible for processing a particular block of information. The central element of the system is key interest rate forecasting, which serves as a basis for subsequent analysis. Within the system architecture, fundamental analysis, technical analysis, credit risk assessment of bonds, and an adapted classical portfolio optimization model are integrated, ensuring consistency of recommendations across different asset classes.

Recent advances in artificial intelligence and digital financial architectures, including metaverse-oriented hypernetwork and blockchain-based systems Ruponen et al. (2025), highlight the growing role of intelligent, modular decision-support systems in finance.

The proposed approach is aimed at reducing the time lag between changes in macroeconomic conditions and the updating of investment decisions. In addition, the probabilistic nature of the system's outputs makes it possible to implement quantitative risk management and scenario analysis mechanisms.

## 2    PROBLEM STATEMENT AND LITERATURE REVIEW

Institutional and retail investors are increasingly faced with the need for adaptive portfolio management in a dynamically changing macroeconomic environment. Heightened interest rate volatility, inflation fluctuations, as well as external and internal economic shocks, call for a shift from static portfolio selection models to flexible multicriteria frameworks capable of responding promptly to changes in external conditions.

Classical approaches, including the Markowitz portfolio optimization model Markowitz (1952), provide a foundation for asset allocation but have a number of limitations related to the need for manual parameter recalculation, insufficient sensitivity to central bank policy, and the omission of qualitative characteristics of issuers. The development of digital technologies and machine learning methods has created prerequisites for integrating intelligent systems into investment analysis processes.

In the international literature, the concept of using neural networks for forecasting macroeconomic indicators and asset valuation has received considerable attention Atsalakis & Valavanis (2009); Fischer & Krauss (2018); Krollner et al. (2010). In particular, convolutional and recurrent neural networks are employed for time series analysis of financial data, which makes it possible to improve the accuracy of short-term market trend forecasts. The studies in Sezer et al. (2017); Kubo & Nakagawa (2025) also examine models that combine machine learning with fundamental valuation of equities and bonds. However, approaches to integrating these tools into a unified architecture that explicitly accounts for monetary policy as a pervasive decision-making factor remain underdeveloped.

In parallel, there is growing interest in broader AI-driven and "digital twin"-like frameworks for financial and economic systems, including metaverse architectures based on hypernetworks and blockchain synergy Ruponen et al. (2025), which conceptually motivate the design of modular, data-driven decision-support platforms.

In the Russian market, research focused on building end-to-end adaptive asset management systems is still at an early stage. Despite the existence of publications devoted to technical analysis, fundamental ratios, and specific aspects of risk management, comprehensive models that combine macroeconomic forecasting and portfolio optimization are extremely scarce.

Thus, there is a pressing need to develop a methodology that integrates macroeconomic, fundamental, technical, and credit factors into a unified investment decision-making system capable of adapting to changes in monetary policy parameters.

## 3    RESEARCH AIM AND OBJECTIVES

The aim of this research is to develop a methodology for forming an adaptive investment portfolio structure based on the integration of neural network models that process macroeconomic, fundamental, technical, and credit factors within a unified investment decision-making system.

In line with this aim, the following objectives are addressed:

Table 1: Input data for the key rate module

| Date | Key rate | Inflation | USD rate | Obs. infl. | Exp. infl. | ... |
|------|----------|-----------|----------|------------|------------|-----|
| 01.01.14 | 5.5 | 6.07 | 33.46 | 6.21 | 14.24 | ... |
| 01.02.14 | 5.5 | 6.21 | 35.22 | 6.14 | 13.33 | ... |
| 01.03.14 | 7.0 | 6.92 | 36.21 | 5.55 | 13.23 | ... |
| 01.04.14 | 7.5 | 7.33 | 35.66 | 6.31 | 13.10 | ... |
| 01.05.14 | 7.5 | 7.59 | 34.93 | 4.33 | 12.18 | ... |
| ... | ... | ... | ... | ... | ... | ... |

1. To substantiate the need for constructing a multi-module investment analysis architecture that explicitly accounts for the role of the interest rate as a key macroeconomic parameter.

2. To develop a system structure that includes specialized neural network modules for:

   - forecasting the key interest rate;
   - fundamental valuation of equities;
   - technical analysis of market data;
   - credit risk assessment of debt instruments;
   - portfolio optimization based on the Markowitz model.

3. To design a mechanism for integrating the modules within a digital twin that ensures consistency of output recommendations.

4. To implement an adaptive mechanism for updating model parameters depending on scenarios of changes in macroeconomic conditions.

5. To assess the effectiveness of the system based on theoretical scenario analysis that takes into account the dynamics of the key rate and market parameters.

6. To identify directions for further research, including data processing automation, expansion of the asset universe, and integration of behavioral and multi-agent factors.

## 4 METHODOLOGY (MODEL ARCHITECTURE, NEURAL NETWORK MODULES, INTEGRATION LOGIC)

The developed system for forming an adaptive investment portfolio structure integrates five specialized analytical modules built on neural network algorithms. All modules are trained on historical data for the period 2014–2026, covering key macroeconomic indicators, market data, corporate reporting, and the characteristics of debt instruments.

### 4.1 KEY RATE FORECASTING

The first module, responsible for forecasting the Bank of Russia key rate, is implemented as a univariate variant of the Temporal Fusion Transformer (TFT) architecture built using the PyTorch Forecasting library and trained on monthly macroeconomic data from January 2014 to December 2025. The input comprises 23 macroeconomic indicators, including the inflation rate, ruble exchange rate, labour market indicators, credit volumes, inflation expectations, and monetary aggregates. These data are combined into a single table and reflect the main blocks of the economy used in the Bank of Russia's modelling framework (external conditions, real sector, macroeconomic indicators, and monetary conditions). Table 1 below shows an abbreviated example of input data.

Formally, we define the target variable as the monthly change in the policy rate,

$$y_t = r_t - r_{t-1},$$

where $r_t$ denotes the key rate in month $t$ and $r_{t-1}$ its value in the previous month. Thus, $y_t$ is positive when the rate is increased, negative when it is cut, and zero when it remains unchanged. The input covariate vector $x_t$ collects all macroeconomic indicators observed at time $t$.

Table 2: Stock valuation input data

| Ticker | Year | Revenue | Op. profit | EBITDA | CAPEX | FCF ... |
|--------|------|---------|------------|--------|-------|---------|
| AFLT | 2014 | 319.8 | 11.3 | 24.8 | 6.03 | 17.5 ... |
| AFLT | 2015 | 415.2 | 44.1 | 58.7 | 8.57 | 30.9 ... |
| AFLT | 2016 | 495.9 | 63.3 | 78.0 | 10.2 | 84.0 ... |
| AFLT | 2017 | 532.9 | 40.4 | 56.0 | 7.68 | 61.8 ... |
| ... | ... | ... | ... | ... | ... | ... |

The TFT module predicts conditional quantiles $\hat{y}_t^{(q)}$ for several levels $q$, and its parameters are estimated by minimising the standard quantile (pinball) loss

$$L(\theta) = \sum_t \sum_{q \in Q} \rho_q\big(y_t - \hat{y}_t^{(q)}\big), \quad \rho_q(u) = \big(q - \mathbf{1}_{\{u<0\}}\big)u,$$

where $u = y_t - \hat{y}_t^{(q)}$ denotes the forecast error for quantile level $q$. This loss penalises underprediction and overprediction asymmetrically, so that the model learns to place $\hat{y}_t^{(q)}$ at the desired quantile of the conditional distribution of $\Delta r_t$. Based on the predicted quantiles, the conditional distribution of $\Delta r_t$ is approximated and partitioned into five bands corresponding to the scenarios of "substantial decrease", "moderate decrease", "unchanged", "moderate increase" and "substantial increase"; integrating the estimated distribution over these bands yields the scenario probabilities reported by the model. The one-month decision recommendation is obtained by converting the median forecast into a discrete increase/hold/decrease classification using a fixed step size in the key rate.

### 4.2 FUNDAMENTAL EQUITY ANALYSIS

The second module provides fundamental equity valuation taking into account macroeconomic forecasts, corporate financial metrics, market capitalization, profitability indicators, debt structure, credit ratings, dividend yield, and the information background. The inputs are financial and macroeconomic indicators for year $t$, while the outputs are a forecast of annual return and its classification into one of five recommendation types (from "strong sell" to "strong buy"). Table 2 shows a shortened version of the module's input data.

Internally, this is implemented as a multilayer neural network that first learns a generalized representation of the company's state under specified macro conditions and then uses it to estimate both continuous return and the recommendation class. Concretely, for issuer $i$ and reporting year $t$, the model takes a feature vector $x_{i,t}$ constructed from financial and macroeconomic indicators, including the average key rate, and outputs a scalar prediction of one-year log-return $\hat{r}_{i,t}$ together with logits $\hat{z}_{i,t} \in \mathbb{R}^5$ over five recommendation classes from "strong sell" to "strong buy". Logits $\hat{z}_{i,t}$ are unconstrained real numbers that are later converted to class probabilities via the softmax transformation.

The parameters are trained by minimising a weighted sum of a regression term and a multiclass classification term,

$$L(\theta) = \sum_{i,t} \Big(\lambda_{\mathrm{reg}} \big(r_{i,t} - \hat{r}_{i,t}\big)^2 + \lambda_{\mathrm{cls}} \, \mathrm{CE}(y_{i,t}, \hat{z}_{i,t})\Big),$$

where $r_{i,t}$ is the realised one-year log-return, $y_{i,t}$ is the observed recommendation label and CE denotes the multiclass cross-entropy loss. The hyperparameters $\lambda_{\mathrm{reg}}$ and $\lambda_{\mathrm{cls}}$ control the relative importance of accurate return prediction versus accurate recommendation classification.

At the application stage, the fair one-year-ahead price is computed as

$$\hat{P}_{i,t+1} = P_{i,t} \, \exp\big(\hat{r}_{i,t}\big),$$

and the upside

$$\frac{\hat{P}_{i,t+1} - P_{i,t}}{P_{i,t}}$$

is mapped to the final textual recommendation via fixed thresholds (e.g. "strong buy" for expected growth above 15% and "strong sell" for expected decline exceeding 10%), as already described in

Table 3: Input data of the technical analysis module

| Ticker | Date | Price | Open | High | Low | Volume |
|--------|------|-------|------|------|-----|--------|
| SBER | 06.01.2014 | 98.91 | 100.20 | 100.31 | 98.62 | 31690000 |
| SBER | 08.01.2014 | 98.19 | 99.10 | 99.41 | 97.85 | 42370000 |
| SBER | 09.01.2014 | 98.00 | 98.44 | 98.77 | 97.69 | 45990000 |
| ... | ... | ... | ... | ... | ... | ... |

this subsection. Using log-returns ensures that the mapping from $\hat{r}_{i,t}$ to prices is multiplicative and time-consistent, which is standard in financial applications.

## 4.3 TECHNICAL ANALYSIS

The third module performs technical analysis of equities and generates probabilistic signals for price direction over three horizons (7, 30, and 180 days). The module's input data include basic market variables: closing price, opening price, high, low, and trading volume for liquid stocks listed on the Moscow Exchange. Data are updated using an automated module for downloading and refreshing price series via the Information and Statistical Server (ISS) of MOEX and the `apimoex` library for Python. Table 3 shows a shortened version of the module's input data.

On this basis, a unified feature set is constructed: logarithmic returns at several lags, trend indicators (SMA, EMA with different windows, and spreads between moving averages), volatility indicators (historical volatility, ATR, Bollinger bands and their width), oscillators (RSI, stochastic oscillator, CCI, Williams %R), directional movement indicators (ADX, +DI, –DI, Aroon Up/Down), volume-based indicators (volume z-score, OBV), and binary candlestick patterns.

Let $x_{i,t}$ denote the feature vector for stock $i$ on date $t$, comprising the technical indicators listed above. The MLP outputs three real-valued scores $\hat{z}_{i,t}^{(7)}, \hat{z}_{i,t}^{(30)}, \hat{z}_{i,t}^{(180)}$, which are transformed into probabilities of price increase over 7, 30 and 180 days via the logistic link

$$\hat{p}_{i,t}^{(H)} = \frac{1}{1 + \exp\left(-\hat{z}_{i,t}^{(H)}\right)}, \quad H \in \{7, 30, 180\}.$$

Here $\hat{p}_{i,t}^{(H)}$ represents the model-estimated probability that the $H$-day forward return for stock $i$ on date $t$ is positive. During training, the parameters are fitted by minimising the sum of binary cross-entropy losses across horizons, using labels that indicate whether the realised $H$-day return is positive (1) or non-positive (0).

In the trading rules described in Section 5, the short-horizon score

$$S_{i,t}^{(7)} = 2\,\hat{p}_{i,t}^{(7)} - 1$$

serves as the primary signal, while the 30- and 180-day probabilities act as consistency filters and scaling factors for position weights. Thus, $S_{i,t}^{(7)}$ ranges from $-1$ (strong downside) to $+1$ (strong upside), with 0 corresponding to a neutral signal.

## 4.4 CREDIT RISK ASSESSMENT OF BONDS

The fourth module analyses bonds, including issuer solvency, duration, credit rating, spreads, and default probabilities. At the first stage, an automatic download and update of the bond reference database is performed using Moscow Exchange data. A complete list of outstanding issues is formed with identifiers, instrument type (sovereign, sub-federal, corporate, municipal), coupon parameters (fixed, floating, discount bonds), nominal value, currency, and maturity date. Based on these data and current market quotes, key market characteristics are calculated. Table 4 shows a shortened version of the module's input data.

The second block of the module is responsible for integrating credit and fundamental analysis of the issuer. For each issuer, a database of financial statements and ratings is used, including operating profit, EBITDA, free cash flow, debt and net debt, interest expenses, assigned credit rating, and the initial one-year point-in-time default probability. From these data, aggregate covenant indicators are

Table 4: Input data of the bonds module

| SECID | ISIN | SHORTNAME ... |
|---|---|---|
| RU000A0JWDN6 | RU000A0JWDN6 | RUSALBrB01 ... |
| RU000A0JWHW8 | RU000A0JWHW8 | Novsib 8ob ... |
| RU000A0JWRV9 | RU000A0JWRV9 | MTS BO-02 ... |
| ... | ... | ... ... |

derived: the net-debt-to-EBITDA ratio as a measure of leverage, the interest coverage ratio (ICR), and the magnitude of free cash flow. The initial default probability obtained from the rating model is then adjusted depending on these covenants. In the presence of low interest coverage, elevated leverage, and negative free cash flow, the default probability is scaled upward, thereby reflecting a weak financial profile over and above the formal rating. In this way, an adjusted one-year default probability is obtained, which is subsequently used as one of the key inputs.

For feature engineering, a unified tabular dataset is constructed in which each row corresponds to a specific bond issue as of the valuation date. A multilayer fully connected neural network (MLP) is trained on these features to perform multiclass classification of bond investment attractiveness. The target variable is the recommendation produced by the original module: "strong buy", "buy", "hold", "sell", or "strong sell". Formally, for each bond issue $j$ with feature vector $x_j$, the MLP outputs logits $\hat{z}_j \in \mathbb{R}^5$ over the five recommendation classes, which are converted into class probabilities via the softmax function

$$\hat{p}_{j,k} = \frac{\exp(\hat{z}_{j,k})}{\sum_{\ell=1}^5 \exp(\hat{z}_{j,\ell})}, \quad k = 1, \ldots, 5.$$

The model is trained by minimising the multiclass cross-entropy loss

$$L(\theta) = -\sum_j \sum_{k=1}^5 \mathbf{1}\{y_j = k\} \log \hat{p}_{j,k},$$

where $y_j$ is the class assigned by the original rule-based module. This procedure encourages the network to assign high probability to the recommendation class prescribed by the initial expert-based system. For interpretability, we also consider a scalar score defined as

$$S_j = \hat{p}_{j,\text{strong buy}} - \hat{p}_{j,\text{strong sell}},$$

which summarises the balance of evidence in favour of or against a given bond. Values of $S_j$ close to $+1$ indicate a strong buy signal, while values close to $-1$ indicate a strong sell signal.

## 4.5 ADAPTIVE PORTFOLIO OPTIMIZATION

The fifth module implements the mean-variance Markowitz optimization model. The proposed module realizes an adaptive portfolio formation strategy based on the Markowitz framework, where asset class allocation and security selection depend on four inputs: key rate forecasts, fundamental equity analysis, bond analysis, and technical analysis. First, using interest rate scenarios and fundamental and technical signals, the target shares of equities, federal government bonds (OFZ), sub-federal bonds, and corporate bonds are determined: under high rates and expectations of further rate increases, the allocation shifts towards floaters, OFZ, and sub-federal bonds; under expectations of rate cuts and strong equity signals, the focus shifts towards equities and higher-risk bonds.

Then, within each asset class, a separate Markowitz optimization problem is solved using expected returns (for equities, derived from fundamental upside adjusted for short- and medium-term technical signals; for bonds, from yield to maturity adjusted for the expected key rate), subject to constraints on maximum weight per security and per issuer, as well as minimum position size. In formal terms, for each asset class $c$ we consider a vector of expected returns $\mu^{(c)}$ and a covariance matrix $\Sigma^{(c)}$, and we determine intra-class weights $w^{(c)}$ by solving a constrained mean–variance problem that maximises a risk-adjusted objective under non-negativity and concentration constraints. At the class level, target shares for equities, federal government bonds, sub-federal and corporate bonds are computed as a function of the current and forecasted key rate and the aggregate signals from

the fundamental and technical modules; these shares are then used to rescale the intra-class weights, and issuer-level caps and minimum position sizes are applied before renormalisation.

At the second layer, expected returns and risks are computed for the resulting asset set and passed to the Markowitz module. Expected returns are estimated over a 252-trading-day window and further scaled according to the statistical significance of the trend via the t-statistic: expectations are amplified for assets with a persistent positive trend and remain close to baseline for assets without a statistically significant trend. The covariance matrix is estimated from the same historical data. In the empirical implementation, this scaling is performed by multiplying the sample mean of each asset by a function of its t-statistic over the 252-day window, with t-statistics clipped to a finite range. The resulting vector of adjusted expected returns $\tilde{\mu}_t$ and covariance matrix $\Sigma_t$ are then used to define a Sharpe-maximisation problem

$$\max_{w} \frac{\tilde{\mu}_t^\top w}{\sqrt{w^\top \Sigma_t w}}$$

subject to

$$\mathbf{1}^\top w = 1, \quad w \geq 0,$$

with additional explicit limits on the aggregate share of risky assets and an adjustment for transaction costs proportional to portfolio turnover at each rebalancing date. Here $w$ denotes the vector of portfolio weights, $\mathbf{1}$ is a vector of ones, and the constraints enforce full investment, long-only positions, and risk and concentration limits in the spirit of the Markowitz mean–variance framework.

The resulting weight vector defines the target portfolio composition for the next rebalancing period.

On top of this explicitly defined algorithm, the digital twin incorporates a compact machine learning component responsible for calibrating its parameters to different market regimes. Unlike the neural extension of the Markowitz module described in Subsection 4.5, this calibration layer does not solve an additional optimization problem or generate its own portfolio weights; instead, it adjusts the hyperparameters of the existing scheme (the threshold for technical signals, the maximum admissible number of stocks, the amplification coefficients for expected returns based on the t-statistic, allowable risk levels and related configuration parameters) using historical data. A portfolio-level performance criterion that combines return and risk (for example, the Sharpe ratio or its drawdown-adjusted variants) is used as the objective, and the optimisation is carried out with respect to these hyperparameters rather than individual asset-level forecasts.

In the resulting setup, security selection is determined by explicit threshold rules and filters based on fundamental and credit characteristics, while portfolio weights are computed by solving a classical Markowitz mean–variance optimisation problem with calibrated constraints and risk parameters. The machine learning component acts as an adaptation mechanism that tunes these hyperparameters to historical market regimes, without changing the underlying structure of the optimisation problem or the explicit selection and allocation rules.

The final decision on including assets in the portfolio is made by an expert based on the scenario analysis and probabilistic forecasts provided.

## 4.6 IMPLEMENTATION DETAILS AND DATA SIZES

The empirical implementation relies on real data from the Russian market. The key rate forecasting module is trained on a monthly macroeconomic panel covering the period from January 2014 to December 2025 and consists of 144 time steps and 31 input features per observation. The technical analysis module is trained on daily OHLCV data for liquid Moscow Exchange equities; the resulting feature table contains 97439 rows and 7 base market variables per row, which are further transformed into a richer set of derived indicators. The fundamental equity analysis module uses a panel of 254 issuer-year observations with 42 financial and macroeconomic features per row. The bond credit risk module is based on a cross-section of 8208 bond issues, each described by 27 features including instrument characteristics, market quotes and issuer-level financial covenants. These dataset dimensions informed the choice of model architectures and regularisation schemes in each module and reflect that the system is trained on realistic, empirically observed samples rather than purely synthetic data.

The system's architecture and module reconciliation logic are illustrated in Figure 1.

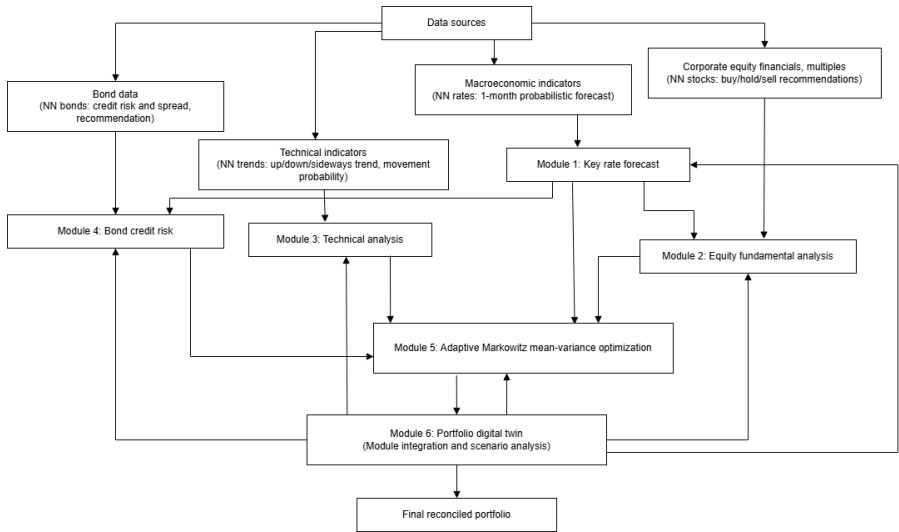

Figure 1: System architecture.

Table 5: Key rate forecast

| id | key_rate_last | key_rate_raw | key_rate_pred | scenario ... |
|---|---|---|---|---|
| cbr_keyrate | 16.0 | 15.57927 | 15.5 | prefer_down ... |
| ... | ... | ... | ... | ... |

## 5   ANALYSIS OF RESULTS

The operation of the proposed system is based on coordinated interaction between specialized modules, each of which processes a specific class of information and generates partial investment recommendations. The final recommendation for each asset is formed from aggregated system outputs that are adapted to current macroeconomic conditions.

At the stage of forecasting monetary policy parameters, probabilistic scenarios for changes in the key policy rate are generated. The latest forecast table illustrates the model's behaviour at the current phase of the monetary policy cycle: it predicted the key rate at 15.5%, which matched the actual decision of the Bank of Russia (15.5%). An example of the output table is shown in Table 5.

Given the observed dynamics of macroeconomic indicators, the model generates a scenario of gradual monetary policy easing. The baseline forecast assumes a smooth reduction of the key rate by about 0.5 percentage points over subsequent meetings.

Backtesting shows that the model reliably identifies the direction of key rate changes, while the exact size of the step becomes less accurate as the forecast horizon increases. Over the short horizon of a single meeting, the mean absolute error is about 0.5 percentage points, which is acceptable for tactical allocation tasks. As the horizon extends to 6–12 meetings, MAE increases from roughly 2.4 to 4.8 percentage points, indicating error accumulation mainly in the amplitude of changes, while the phases and direction of the monetary policy cycle are still captured correctly. However, even the central bank's own medium-term guidance does not provide precise estimates of future policy, so the main focus here is on the one-month forecast and detecting likely short-term shifts in monetary policy.

In the second module, changes in the key rate forecast trigger a reassessment of equity fair values. Scenario analysis implements sensitivity of valuations to five variants of monetary policy: gradual easing, accelerated rate cuts, maintenance of the current level, rate hikes, and accelerated rate hikes. This enables dynamic re-assessment of the investment attractiveness of corporate securities. An example of the module output, taking into account the output of Module 1, is shown in Table 6.

Table 6: Equity recommendations

| Ticker | Year | key_rate_end_12m | rating_label_final ... |
|--------|------|------------------|------------------------|
| AFLT | 2024 | 10 | Strong sell ... |
| ALRS | 2024 | 10 | Strong sell ... |
| BSPB | 2024 | 10 | Sell ... |
| CHMF | 2024 | 10 | Strong sell ... |
| GAZP | 2024 | 10 | Strong buy ... |
| GMKN | 2024 | 10 | Sell ... |
| ... | ... | ... | ... ... |

Table 7: Technical analysis module output

| Ticker | P_Up_H7 | P_Down_H7 | P_Up_H30 | P_Down_H30 | P_Up_H180 | P_Down_H180 |
|--------|---------|-----------|----------|------------|-----------|-------------|
| AFLT | 0.63 | 0.37 | 0.75 | 0.25 | 0.78 | 0.22 |
| ALRS | 0.41 | 0.59 | 0.47 | 0.53 | 0.58 | 0.42 |
| BSPB | 0.49 | 0.51 | 0.53 | 0.47 | 0.62 | 0.38 |
| CHMF | 0.55 | 0.45 | 0.59 | 0.41 | 0.66 | 0.34 |
| GAZP | 0.47 | 0.53 | 0.53 | 0.47 | 0.55 | 0.45 |
| GMKN | 0.57 | 0.43 | 0.61 | 0.39 | 0.58 | 0.42 |
| IRAO | 0.63 | 0.37 | 0.81 | 0.19 | 0.80 | 0.20 |
| LKOH | 0.38 | 0.62 | 0.38 | 0.62 | 0.53 | 0.47 |
| ... | ... | ... | ... ... | | | |

The technical analysis module, using 42 indicators, identifies short-term and medium-term market signals. When technical and fundamental signals diverge, the system flags a conflict and recommends revisiting selected model parameters. Backtesting of the technical analysis module was performed on historical data for financial instruments over the period from 01.01.2014 to 01.01.2026. The module uses 42 technical indicators and probabilistic forecasts over three horizons (H7, H30, H180); the primary source of signals is the short-term H7 horizon, while H30 and H180 serve as filters for directional consistency and for adjusting position weights. Risk management is implemented via volatility-adaptive stop-losses and take-profits based on ATR(14) with a fixed target profit-to-risk ratio of at least three, a maximum holding period of 7 days, and accounting for basic transaction costs.

Over the entire test period, about 2,600 trades were generated, providing a sufficient number of observations for statistical evaluation of the strategy's properties. The cumulative simple return relative to initial capital was approximately 3,072.13%, and the average 7-day trade return was about 2.293%. The approximate Sharpe ratio, calculated from daily changes in the equity curve, was around 6.39, indicating a very high return-to-risk ratio within the studied sample. At the same time, it is emphasized that the backtest results were obtained under simplified assumptions about market infrastructure (limited treatment of transaction costs, no modeling of slippage and liquidity) and a fixed rule set, which requires further robustness checks on alternative samples and in stress scenarios. Preliminary live-market testing with the same rule set indicates that the signals are usable in practice, although their out-of-sample performance remains subject to further monitoring and robustness checks. An example of the module output showing the probability of stock price movement is given in Table 7.

In the bond analysis module, credit risk is assessed, including the calculation of fair spreads over government bonds. For example, in the environment of rising key rates in 2022–2024, the system recommended increasing portfolio duration by purchasing long-term floating-rate bonds. After rate stabilization in 2025–2026, the strategy shifts towards duration reduction. An example of the module output is shown in Table 8.

The empirical evaluation of the digital twin's effectiveness was performed on Russian market data for the period 01.01.2020–01.01.2024, which includes episodes of elevated volatility linked to the 2020 pandemic and the 2022 geopolitical crisis. As a benchmark, the Moscow Exchange Total Return Index (MCFTR) was used. Based on daily index values, monthly levels and realized monthly returns were computed. For comparison, an investor with initial capital of 1 million RUB was

Table 8: Bonds module output

| SHORTNAME | signal | issuer_rating | key_rate_now | key_rate_12m |
|---|---|---|---|---|
| RUSALBrB01 | BUY | A+(RU) | 15.5 | 10 ... |
| Novsib 8ob | SELL | – | 15.5 | 10 ... |
| Udmurt2016 | BUY | – | 15.5 | 10 ... |
| Rostel1P1R | HOLD | AAA(RU) | 15.5 | 10 ... |
| ... | ... | ... | ... | ... ... |

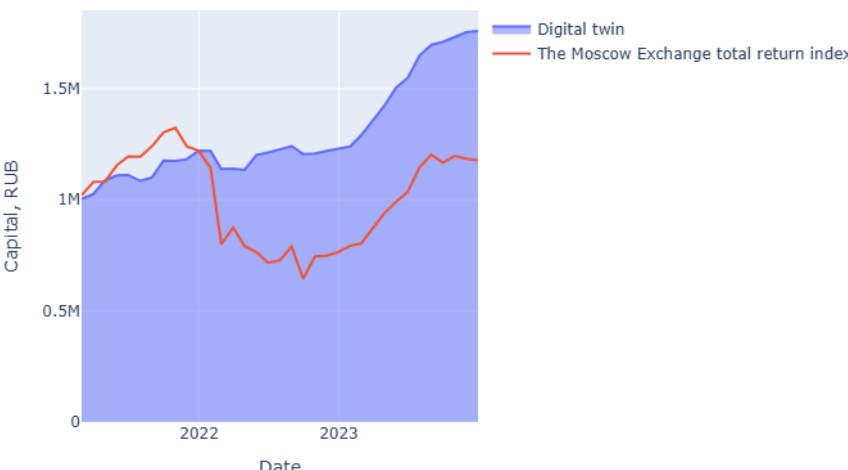

Figure 2: Digital twin and Moscow Exchange total return index.

considered, who either passively tracks the total return index, or uses the trading decisions generated by the digital twin, or applies the classical Markowitz module.

According to the backtest, the total return index over 2020–2023 increased capital from 1 million to about 1.3–1.4 million RUB, corresponding to cumulative growth of roughly 34% and an average annual return close to 10%. The maximum drawdown of the index from the local peak at the end of 2021 to the low in March 2022 was about 28% (from 1.22 million to 876 thousand RUB).

Under the same starting conditions, the portfolio formed by the digital twin increased capital from 1 million to about 1.76 million RUB over the same period. Its average monthly return was about 1.68%, with monthly volatility around 2.75%, which implies an effective average annual return of roughly 21.6%, annual volatility near 9.5%, and an annual Sharpe ratio of 2.27. The maximum drawdown of this portfolio in 2022 did not exceed 8% from the local maximum (from 1.22 million to 1.14 million RUB), which is significantly smaller than the drawdown of the total return index.

Separately, a classical implementation of the Markowitz module was tested, using expected returns and the covariance matrix estimated from historical data, without the additional layer of technical signals. Starting from 1 million RUB, this module produced a capital path ending at about 1.48 million RUB over 01.01.2020–01.01.2024, corresponding to a total return of 48.32% and 35 portfolio rebalancings over the period.

In this backtest, the digital twin, which relies on a richer set of technical signals and dynamic control of exposure to risky assets, achieved higher terminal returns and smaller drawdowns than both passive tracking of the total return index and the baseline Markowitz module.

Figure 2 shows the capital trajectory for an investor with an initial capital of 1 million RUB under passive tracking of the total return index and when using the digital twin.

The chart shows that during market upswings (2020–2021 and 2023) the adaptive portfolio participates in the growth, albeit with a more moderate exposure to risky assets, whereas in the shock period of 2022 the decline in portfolio value is notably smaller than that of the index.

In this configuration, the digital twin aligns the outputs of individual modules within a single optimisation scheme and enforces consistency between fundamental, technical and credit signals by excluding assets that do not meet all required conditions, even if their historical returns appear attractive. On the basis of the harmonised parameters of expected return and risk, the Markowitz model then constructs a portfolio that satisfies the specified risk constraints and limits on the share of risky assets. This setup integrates heterogeneous information sources into a unified decision procedure and allows the portfolio composition to adjust to changes in macroeconomic conditions and market signals.

# 6 JUSTIFICATION FOR CLASSIFYING THE GENERATED PORTFOLIO AS AN ADAPTIVE STRUCTURE

In Khudyakov & Barykin (2026) an adaptive investment portfolio structure is defined as a formalized configuration of assets whose weights are functionally dependent on a vector of market states and macroeconomic signals, enabling the preservation of target performance parameters under changing macroeconomic regimes. A crucial distinction lies in separating the notions of adaptive structure and adaptive strategy: the former refers to an inherent property of the portfolio itself, whereas the latter denotes an external set of management rules.

## 6.1 DYNAMIC ASSET ALLOCATION

The first element of the definition Khudyakov & Barykin (2026) implies that portfolio weights are updated through an event-driven trigger mechanism rather than on a purely calendar-based schedule. In Ivanyuk (2024) it is shown that rebalancing in adaptive systems fundamentally differs from simple restoration of initial weights. In Tinyakova & Chervontseva (2020) this logic is extended by demonstrating that the optimal portfolio structure should distinguish between extensive and intensive return-generation regimes.

In the proposed architecture, the event-driven mechanism is implemented via a cascade of modules: a shift in the forecast key rate, a change in fundamental equity valuation, a revision of technical signals, or a change in credit risk assessment triggers rebalancing. Target weights are re-determined as the solution to an optimization problem with updated inputs.

## 6.2 FORWARD-LOOKING ORIENTATION

The second element of the definition Khudyakov & Barykin (2026) requires the use of forecast, rather than purely historical, parameters in portfolio construction. In Miroshnikov (2019) the concept of predictive optimality is introduced. In the presented system, the central module produces a probabilistic distribution of rate scenarios; the fundamental module estimates fair values under monetary policy scenarios; the technical module evaluates probabilities of price movement; the credit module estimates bond spreads taking into account the expected rate path; and the optimization block operates with forecast returns and covariances.

## 6.3 RESPONSE TO MARKET REGIMES

The third element Khudyakov & Barykin (2026) relates to the response to discrete market regimes. In Ivanyuk (2024) five discrete market states are identified, and an optimal portfolio structure is derived for each. In Segnon et al. (2024) it is shown that portfolios should adapt to changes in volatility based on regime shifts. In the proposed architecture, regime identification is carried out through the probabilistic scenarios produced by the key rate module; regime switching triggers adjustments in bond duration, equity share, and risk limits.

## 6.4 MATHEMATICAL MODEL AND COMPUTATIONAL ARCHITECTURE

The fourth element Khudyakov & Barykin (2026) requires the use of extensions of portfolio theory and modern computational methods. In Konovalova & Abuzov (2024) the effectiveness of combining Markowitz theory with genetic algorithms and LSTM networks is demonstrated. In Kubo & Nakagawa (2025) deep neural networks are employed to model nonlinear relationships between returns and external factors. In the proposed system, the adapted Markowitz optimization model is complemented by an ensemble of neural network modules integrated via the digital twin.

## 6.5 ADAPTIVE RISK MANAGEMENT

The fifth element Khudyakov & Barykin (2026) assumes flexible risk management that adapts to market conditions. . In Ivanyuk (2024) a combined forecast of systematic risk based on price standard deviations is applied, enabling the model to adjust to current volatility. In Segnon et al. (2024) dynamic risk management is emphasized as an integral component of an adaptive structure through the detection of volatility regime shifts. In Waga et al. (2025) develop the idea of robust optimization: instead of minimizing risk under fixed parameters, the system minimizes risk under the worst case plausible scenario, which is particularly relevant under uncertainty. In the proposed system, adaptive risk management is implemented through recalculation of portfolio risk at each update of forecast parameters and a variable allowable risk level that depends on the current macroeconomic regime. When expected volatility rises, the optimization block tightens constraints on the maximum share of risky assets, while stable forecasts allow risk limits to be relaxed. The credit module embeds risk management at the instrument level: fair credit spread to OFZ, default probability, and duration are used as input constraints for optimization. The technical analysis module warns against purchases during pronounced downward price movements.

## 7 CONCLUSIONS AND RECOMMENDATIONS

This study proposes an approach to forming an adaptive investment portfolio structure based on the integration of macroeconomic analysis, fundamental and technical asset valuation, credit risk analysis, and mean-variance optimization. The system is implemented as an ensemble of neural network modules capable of generating probabilistic investment recommendations that reflect current and expected monetary policy parameters.

The practical significance lies in its potential application to both individual portfolio management and corporate investment strategy design. The system reduces the time lag between changes in macroeconomic conditions and portfolio adjustment and establishes a unified analytical framework suitable for the digital transformation of investment activities.

Going forward, the system is planned to be extended to the full spectrum of liquid assets traded on the Moscow Exchange, with automated data updating and further development of the digital twin. Particular attention will be devoted to incorporating behavioral factors, building multi-agent risk assessment models, and introducing elements of self-learning architectures based on modern transformer and recurrent networks. An important practical limitation of the current implementation is the partial dependence on manually curated datasets and the absence of a fully automated pipeline for handling missing or delayed information. In further research, the system is planned to be extended with robust data-ingestion and preprocessing layers, including automated updates of issuer fundamentals, bond reference data and macroeconomic indicators, as well as parsers for PDF reports, news feeds and alternative data sources. These enhancements will allow the digital twin to operate in a more fault-tolerant mode, degrade gracefully when individual modules or data channels are temporarily unavailable, and systematically incorporate new information flows into the adaptive portfolio construction process.

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
